# High-Dose Methylprednisolone Pulses for 3 Days vs. Low-Dose Dexamethasone for 10 Days in Severe, Non-Critical COVID-19: A Retrospective Propensity Score Matched Analysis

**DOI:** 10.3390/jcm10194465

**Published:** 2021-09-28

**Authors:** José María Mora-Luján, Manel Tuells, Abelardo Montero, Francesc Formiga, Narcís A. Homs, Joan Albà-Albalate, Xavier Corbella, Manuel Rubio-Rivas

**Affiliations:** 1Department of Internal Medicine, Bellvitge University Hospital, Bellvitge Biomedical Research Institute-IDIBELL, University of Barcelona, 08907 L’Hospitalet de Llobregat, Barcelona, Spain; mtuells@bellvitgehospital.cat (M.T.); asaez@bellvitgehospital.cat (A.M.); fformiga@bellvitgehospital.cat (F.F.); nhoms@bellvitgehospital.cat (N.A.H.); jalba@bellvitgehospital.cat (J.A.-A.); xcorbella@bellvitgehospital.cat (X.C.); mrubio@bellvitgehospital.cat (M.R.-R.); 2School of Medicine, Universitat Internacional de Catalunya, 08195 Barcelona, Spain

**Keywords:** COVID-19, corticosteroids, methylprednisolone, dexamethasone, mortality

## Abstract

Corticosteroids are largely recommended in patients with severe COVID-19. However, evidence to support high-dose methylprednisolone (MP) pulses is not as robust as that demonstrated for low-dose dexamethasone (DXM) in the RECOVERY trial. This is a retrospective cohort study on severe, non-critically ill patients with COVID-19, comparing 3-day MP pulses ≥ 100 mg/day vs. DXM 6 mg/day for 10 days. The primary outcome was in-hospital mortality, and the secondary outcomes were need of intensive care unit (ICU) admission or invasive mechanical ventilation (IMV). Propensity-score matching (PSM) analysis was applied. From March 2020 to April 2021, a total of 2,284 patients were admitted to our hospital due to severe, non-critically ill COVID-19, and of these, 189 (8.3%) were treated with MP, and 493 (21.6%) with DXM. The results showed that patients receiving MP showed higher in-hospital mortality (31.2% vs. 17.8%, *p* < 0.001), need of ICU admission (29.1% vs. 20.5%, *p* = 0.017), need of IMV (25.9% vs. 13.8, *p* < 0.001), and median hospital length of stay (14 days vs. 11 days, *p* < 0.001). Our results suggest that treatment with low-dose DXM for 10 days is superior to 3 days of high-dose MP pulses in preventing in-hospital mortality and need for ICU admission or IMV in severe, non-critically ill patients with COVID-19.

## 1. Introduction

As of 4 August 2021, more than 200 million people have been infected with SARS-CoV-2 and more than 4 million have died from COVID-19. After 1 year of advances in the understanding of the disease, several risk factors have been recognized [1,2]. These include older age, male gender, certain comorbidities, and phenotypic clusters based on patients’ symptoms [3,4,5,6,7,8,9].

Severe patients present with respiratory distress associated with systemic hyperinflammatory syndrome, the so-called “cytokine storm”. Patients showing this serious host-immune response are at higher risk to transit into the most advanced stage of the disease, requiring intensive care unit (ICU) admission or invasive mechanical ventilation (IMV). The decrease in the lymphocyte count and the elevation of inflammatory parameters such as C-reactive protein (CRP), lactate dehydrogenase (LDH), ferritin and D-dimer are part of the laboratory features that characteristically accompany this inflammatory process and determine a worse prognosis [10,11]. Consequently, immunosuppressive agents are presently the cornerstone of treatment regimens for severe COVID-19.

In this regard, and on the basis of the RECOVERY trial [12], other randomized clinical trials [13,14,15,16,17,18,19], and systematic reviews and meta-analyses [20,21,22], current international clinical guidelines recommend the use of dexamethasone 6 mg/day for 10 days in patients with COVID-19 who are mechanically ventilated or require supplemental oxygen [23,24,25].

In addition to dexamethasone, alternative glucocorticoids such as prednisone, methylprednisolone, or hydrocortisone have also been used in various formulations and doses and for varying durations in patients with COVID-19 [20]. However, to date, evidence to support the use of methylprednisolone and hydrocortisone for the treatment of COVID-19 has still not been as robust as that demonstrated for dexamethasone in the RECOVERY trial.

Our study aimed to compare the effect of high-dose methylprednisolone pulses for 3 days vs. low-dose dexamethasone for 10 days in preventing in-hospital mortality, and need of ICU admission or IMV in severe, non-critically ill patients with COVID-19.

## 2. Materials and Methods

### 2.1. Study Design, Patients and Setting

From March 2020 to April 2021, a retrospective observational investigation was performed in the large cohort of patients with COVID-19 admitted to the Bellvitge University Hospital, a 750 bed tertiary care hospital in Barcelona (Spain). All included patients were collected consecutively and diagnosed by polymerase chain reaction (PCR) or rapid antigenic test for SARS-CoV-2 taken from a nasopharyngeal, sputum, or bronchoalveolar lavage specimen. Data collection for each patient in terms of demographic data, comorbidities, symptoms and signs, laboratory data, treatments, and outcomes were verified by review of institutional electronic medical records.

### 2.2. Inclusion Criteria

The included patients were those admitted to our centre due to SARS-CoV-2 pneumonia; who started receiving either intravenous high-dose methylprednisolone (MP) pulses ≥ 100 mg/day for 3 days or oral/intravenous low-dose dexamethasone (DXM) 6 mg/day for 10 days; within the first 3 days of admission at conventional hospital ward; and before their admission to the ICU or the use of IMV, if needed. In accordance with our hospital protocol, MP or DXM treatment was indicated in all hospitalized patients with COVID-19, at the physician’s discretion, in need of supplemental oxygen and presenting with elevated inflammatory parameters (CRP ≥ 75 mg/L, or LDH ≥ 400 U/L, or Ferritin ≥ 700 mcg/L, or D-dimer ≥ 1000 ng/mL).

### 2.3. Exclusion Criteria

Patients were excluded if an episode of nosocomial infection was detected during admission; MP or DXM was started more than 3 days after hospital admission; high-dose MP pulses continued beyond the first 3 days with progressive dose tapering; or the patient died within the first 24 h after starting MP/DXM treatment.

### 2.4. Treatments Prescribed and Definition of Groups

The cohort was divided into 2 study groups. First, the MP group, which included those patients who had received high-dose MP pulses of ≥100 mg/day for 3 days, without subsequent doses (details of the doses of MP used are available in Appendix A). Second, the DXM group, which included those patients who had received the current gold standard based on the RECOVERY study [12] of DXM 6 mg/day × 10 days. The additional use of antivirals (lopinavir/ritonavir, remdesivir), hydroxychloroquine, azithromycin, immunosuppressive drugs such as tocilizumab (TCZ), or low molecular weight heparins (LMWH) were allowed in accordance with the recommended hospital standard of care (SOC) protocol for COVID-19 available in each period of the pandemic.

### 2.5. Outcomes

The primary outcome of this study was in-hospital mortality. The secondary outcomes were transfer to ICU or need for IMV. Furthermore, the length of hospital stay, and the composite variable of in-hospital mortality, requirement of high-flow nasal cannula (HFNC), non-invasive mechanical ventilation (NIMV), invasive mechanical ventilation (IMV), and ICU admission were also assessed.

### 2.6. Statistical Analysis

Multiple imputations of missing data were performed (details in Appendix A). In order to minimize differences between groups and improve comparability, propensity-score matching (PSM) was performed. The PSM included sociodemographic variables (age, sex, and race), days from symptom onset to hospital admission, comorbidities (smoking behavior, body mass index or BMI, arterial hypertension, diabetes mellitus, dyslipidemia, ischemic heart disease, cerebrovascular disease, chronic liver disease, severe chronic renal failure, chronic obstructive pulmonary disease or COPD, asthma, obstructive sleep apnea syndrome or OSAS, chronic heart failure, cancer, dementia, degree of dependency, and Charlson Index), the respiratory rate upon admission, laboratory variables at admission (PaO_2_/FiO_2_, ferritin, lactate dehydrogenase or LDH, C-reactive protein or CRP, lymphocyte count, and D-dimer), and the treatments used during admission in addition to DXM or MP (TCZ and remdesivir). A logistic-regression propensity score nearest neighbor matching (PSM) with replacement and caliper 0.2 was performed.

Categorical variables were expressed as absolute numbers and percentages. Continuous variables were expressed as mean plus standard deviation (SD) in case of parametric distribution or median and interquartile range (IQR) in the case of non-parametric distribution. Differences between groups were assessed using the chi-square test for categorical variables and T–test or Mann–Whitney test as appropriate for continuous variables. *p*-values < 0.05 indicated statistical significance.

To study the risk factors for the primary and the secondary outcomes, binary logistic regression was performed. Those variables with *p* < 0.010 in the univariate study plus age and sex were introduced in the multivariate model. In-hospital mortality was shown graphically using Kaplan–Meier curves. Statistical analysis was performed by IBM SPSS Statistics for Windows, Version 26.0. Armonk, NY: IBM Corp.

## 3. Results

### 3.1. Demographic, Comorbidity and Clinical Data

From March 2020 to April 2021, a total of 2284 patients were admitted to our hospital due to severe, non-critical COVID-19 and were included in our Registry. Of these, 1045 (45.8%) patients were treated with corticosteroids (CS): 583 (25.5%) with DXM 6 mg/day for 10 days, 310 (13.6%) with 3-day MP ≥100 mg/day pulses, and 152 (6.7%) with 3-day MP ≥100 mg/day and progressive dose tapered. Finally, 493 (21.6%) patients of the DXM group and 189 (8.3%) of the 3-day MP group met the inclusion criteria for the present study (Figure 1).

Table 1 shows the comparison of the demographic and comorbidity data between MP and DXM groups. The median age was similar between groups, with a significant predominance of males in the MP group, being Caucasian in the majority of the overall included patients. There was a small difference between groups in the days between the disease onset and the hospital admission (MP 8 vs. DXM 7 days, *p* = 0.011). The main comorbidities were arterial hypertension, dyslipidemia and diabetes mellitus.

Table 2 shows the comparison of symptoms and signs between groups at the time of hospital admission. There were no significant differences between groups except for a greater presence of cough and fever in the MP group.

### 3.2. Laboratory Tests

Table 3 shows the oxygenation (PaO_2_/FiO_2_) and the inflammation parameters (lymphopenia, CRP, LDH, ferritin, D-dimer) between groups. Compared with the DXM group, MP patients presented worse data in all the above-mentioned analytical items. After matching, the 2 groups were homogeneous in all inflammatory parameters except for CRP, which was slightly higher in the MP group.

### 3.3. Additional Treatments

Table 4 shows the concomitant drugs administered in addition to CS regimens, either MP or DXM. The greater use of TCZ and LMWH at high doses in the MP group was noteworthy. Of particular interest was the higher use of remdesivir in the DXM group. To improve comparability, both TCZ and remdesivir were included in the PSM. Despite matching, TCZ remained more used as additional treatment in the MP group.

### 3.4. Primary and Secondary Outcomes

Table 5 shows the comparison of the outcomes between groups. In this respect, in-hospital mortality as the primary outcome and the secondary outcomes of transfer to ICU or need for IMV, the composite variable (in-hospital mortality, requirement of HFNC, NIMV, IMV and ICU admission), and the median hospital length of stay were significantly worse in the MP group.

Analyzing these data after PSM, identical results were obtained, showing significantly worse clinical outcomes in the MP group when compared with the DXM group. Figure 2 depicts the survival between groups.

### 3.5. Risk Factors for In-Hospital Mortality and the Combined Variable

After the multivariate analysis (Table 6), the risk factors for the primary outcome (in-hospital mortality) were older age, higher degree of dependency, lower PaO_2_/FiO_2_ and higher LDH on admission, and to be treated with high-dose MP pulses for 3 days rather than low-dose DXM for 10 days. Regarding the factors associated with worse prognosis measured by the composite variable were lower PaO_2_/FiO_2_, tachypnoea >20 bpm, and higher LDH upon admission, and the use of TCZ in addition to MP/DXM treatment (Appendix A). Conversely, the use of remdesivir was found to be protective in terms of the composite endpoint. Treatment with high-dose MP pulses was found to confer a worse prognosis in the univariate analysis but its effect was lost in the multivariate study.

## 4. Discussion

Corticosteroids are now considered a first-line treatment for hospitalized patients with COVID-19 and systemic hyperinflammatory syndrome. Current evidence largely supports the use of low-dose DXM, as demonstrated in the RECOVERY trial, but there are poorer high-quality data on alternative glucocorticoids used in various formulations and doses and for varying durations.

The present study is one of the largest retrospective high-quality observational studies to compare the use of high-doses MP pulses for 3 days vs. the current gold standard of DXM 6 mg per day for 10 days, in severe, non-critically ill patients with COVID-19. The inclusion criteria were restrictive enough to create 2 groups as homogeneous as possible. Furthermore, a PSM analysis was performed to reduce the bias due to potential confounding variables that could be found in an estimate of the treatment effect obtained from simply comparing outcomes among patients treated with each of the compared arms. Results showed higher beneficial effect of 6 mg DXM for 10 days, compared with high-dose MP for 3 days, in preventing in-hospital mortality, need to ICU admission or IVM.

Our results are in keeping with those observed in other clinical trials and meta-analyses [12,14,15,18,21,22,26,27]. Mortality observed in our DXM arm (17.8%) was slightly lower than that observed in different studies of reference: 22.9% in the RECOVERY trial (DXM 6 mg/day for 10 days), 37.5% in the METCOVID study (MP 0.5 mg twice daily for 5 days), and 32.7% in the REACT Working group meta-analysis (different low-dose of CS regimens). Furthermore, the GLUCOCOVID trial, in which intermediate doses of MP were used, found a 35% rate of mortality when the combined variable of in-hospital mortality, ICU admission, or IMV was assessed. In the CoDEX trial, a multicenter, randomized, open-label, clinical trial conducted in 41 ICUs in Brazil to determine whether intravenous high-dose DXM (20 mg of DXM intravenously daily for 5 days, 10 mg of DXM daily for 5 days or until ICU discharge, plus SOC) increased the number of ventilator-free days among patients with COVID-19-associated acute respiratory distress syndrome (ARDS), patients in the DXM group had significantly lower cumulative probability of having died or being mechanically ventilated at day 15 than the SOC group (67.5% vs. 80.4%; OR, 0.46; 95% CI, 0.26 to 0.81; *p* = 0.01). Other meta-analyses also showed lower in-hospital mortality or need for IMV in patients treated with CS, compared with non-CS treated patients [22,26]. Finally, the REMAP-CAP trial, which used hydrocortisone (HC) at high doses, showed a Bayesian probability superior to the placebo group, although it was a 7-day treatment, not compared with low-dose CS regimen [13].

In contrast, other studies disagree with the present results, most of them carried out at the ICU setting. The trial by Dequin et al. found 14.7% of in-hospital mortality in a cohort of 76 ICU patients receiving low-dose HC [16]. The trial by Edalatifard et al. also found 5.9% of in-hospital mortality with the use of high-doses of MP pulses of 250 mg/d in a group of 34 patients [17]. The retrospective PSM analysis by Rodriguez-Baño et al. reported a 21-day mortality rate of 10.3% in 78 patients receiving high-dose CS pulses (≥250 mg of MP or equivalent) vs. 11.9% in 344 patients of the control group (PSM HR 0.74; 0.31–1.77) [28]. Of note, in the same study, the group of 117 patients treated with CS at intermediate-high doses of CS showed higher mortality compared with controls (18.8%; HR 1.16; 0.66–2.03), although the duration of CS treatment was not specified. Among various retrospective studies comparing different doses of CS, Garcia Muñoz et al. compared 97 patients with at least two MP pulses at 125–250mg/day vs. 30 patients receiving at least two doses of DXM 6 mg/day, showing a need for ICU admission of 12.4% and 30%, respectively [29]. However, a PSM analysis was not performed and the final duration of CS was also not specified. Furthermore, López-Zúñiga et al. retrospectively compared high dose CS (>1.5 mg/kg/day of MP or DXM equivalent) vs. low dose (<1.5 mg/kg/day of MP or DXM equivalent), showing a reduction in mortality with the use of high-dose CS pulse therapy (HR = 0.087; 0.02–0.36; *p* < 0.001) [30]. However, despite the notable sample size (299 severe-critical COVID-19 patients), PSM analysis was not performed, nor did it specify treatment duration or inflammatory parameters at admission. In addition, high dose CS pulse therapy was defined as daily dose of at least 1.5 mg/kg/24 h of MP or DXM equivalent, which might be lower than those used in the present study in the MP group. Finally, Ruiz-Irastorza et al. compared a retrospective sample of 61 patients receiving 3-day MP pulses at week 2 after the symptoms onset vs. 33 receiving out-of-week-2-MP or non-pulses CS (<100 mg/kg/day) vs. 148 who did not receive MP. Results showed an in-hospital mortality rate of 6.6% vs. 9.9% vs. 9.9%, respectively [31]. However, PSM analysis was not performed and the duration of CS was not specified in the non-pulses CS group.

In the present study, we observed outcome differences from the fourth day after the start of treatment, which suggests that the protective effect of CS is not so much dose-dependent as time-dependent. Thus, a low-dose of DXM administered for 10 days would protect against the progression into respiratory distress, instead of administering a high-dose of MP over only three days. It is well known that the immunomodulatory effect of CS is via two mechanisms, genomic and non-genomic. The genomic mechanism is activated at low-intermediate doses and generates a trans-repression effect (decreased production of nuclear factor k-light-chain-enhancer of activated B cells and proinflammatory cytokines) and a trans-activation effect (increased expression of anti-inflammatory molecules) [21,32,33]. On the other side, the non-genomic mechanism is activated at high doses of CS and triggers an activation of intracellular kinase-mediated signaling cascades with anti-inflammatory effect [21,31,34]. Our results suggest that the genomic mechanism of action, activated by low doses of CS, plays a primary role in the COVID-19-activated cytokine storm, over the non-genomic mechanism.

Considering the present results and the above-mentioned studies, some other CS regimens could be tested and compared with the currently recommended low-dose DXM for 10 days to further elucidate which is the most effective CS regimen in severe COVID-19. From our study, it does seem clear that the duration should be a minimum of 10 days. We believe that the next step should be to compare 6 mg DXM × 10 days with a high-dose MP pulses × 3 days followed by a progressive dose tapering. Unfortunately, we could not carry out this comparison because the heterogenicity of this subpopulation in our cohort. This is an opportunity that other inflammatory/autoimmune diseases have not had to date, so there is a window of opportunity in this regard that should be taken advantage of in patients with severe COVID-19.

Our results should be taken with caution for several reasons. Firstly, the study is observational and retrospective in nature, using real-world data. Subsequently, PSM was used to reduce the possibility of bias between MP and DXM outcomes caused by a factor that determined one or the other treatment rather than the treatment itself. Secondly, although the study design accurately selected those patients with marked inflammation, it did not allow to identify those with ARDS. It is well known that some COVID-19 patients, despite being early admitted with initial pneumonia and high inflammatory parameters, have adequate SatO_2_ with low supplementary FiO_2_, since lung damage occurs in the following days. Therefore, although PaO_2_/FiO_2_ was well matched between groups, this might not translate into what degree of ARDS the patient was in. Third, despite matching by propensity score technique, the MP pulses group frequently showed more fever, higher CRP and more concomitant use of TCZ administration when compared with the DXM group. This would mean that patients receiving MP pulses might be in a more severe stage of the disease despite our attempt to appropriately match the two groups. Finally, the study population was predominantly Caucasian, so the conclusions should be taken with caution when applied to other subpopulations. Authors should discuss the results and how they can be interpreted from the perspective of previous studies and of the working hypotheses. The findings and their implications should be discussed in the broadest context possible. Future research directions may also be highlighted.

## 5. Conclusions

Although randomized controlled trials (RCTs) are essential in determining causal relationships between treatment and outcome, high-quality observational studies based on real-world data can fill certain evidence gaps particularly where RCTs have not been conducted or were still ongoing. In this respect, our study suggests that treatment with low-dose DXM for 10 days is superior to 3-day high-dose MP pulses in preventing worse clinical outcomes such as in-hospital mortality and need for both ICU admission and use of IMV, in severe, non-critically-ill patients with COVID-19.

## Figures and Tables

**Figure 1 jcm-10-04465-f001:**
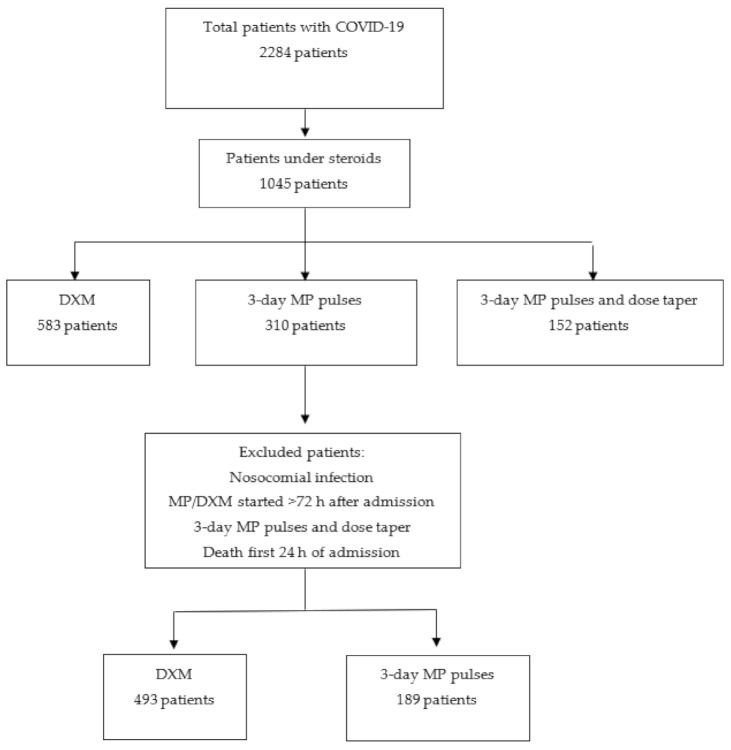
Flow-Chart. DXM: dexamethasone. MP: methylprednisolone.

**Figure 2 jcm-10-04465-f002:**
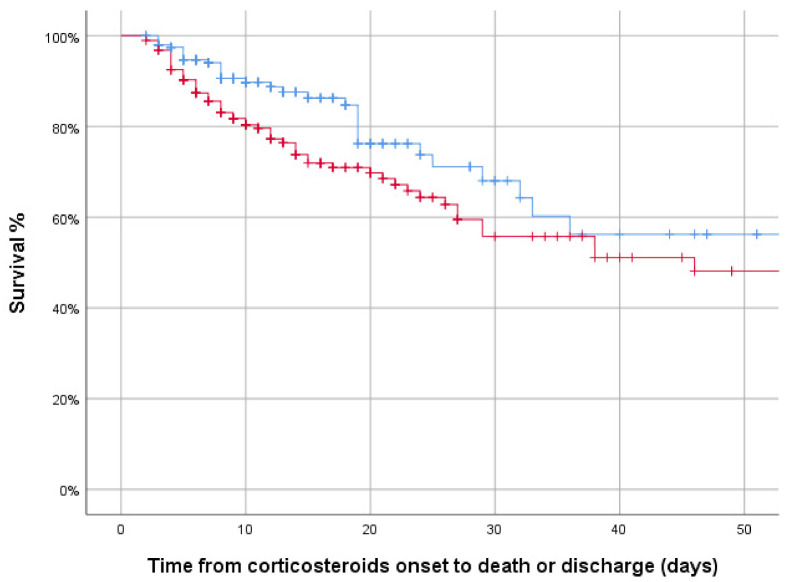
In-hospital mortality between groups in the matched sample: 3-day MP pulses (red) vs. DXM (blue). Kaplan–Meier curves. Log-rank test 2.849 *p* = 0.091.

**Table 1 jcm-10-04465-t001:** Comparison of demographic and comorbidity data between groups.

	Whole Cohort			Matched Sample		
	3-Day MP	DXM	*p*-Value	3-Day MP	DXM	*p*-Value
*n*	189	493		189	199	
Age, median (IQR)	67 (56.6–76.5)	66.6 (56.4–76.1)	0.642	67 (56.6–76.5)	67.8 (56.8–76.6)	0.805
Gender (males)	140 (74.1)	326 (66.1)	0.046	140 (74.1)	143 (71.9)	0.624
Race			0.322			0.879
Caucasian	156 (82.5)	390 (79.1)		156 (82.5)	163 (81.9)	
Hispanic	27 (14.3)	70 (14.2)		27 (14.3)	30 (15.1)	
Others	6 (3.2)	33 (6.7)		6 (3.2)	6 (3)	
Days from onset to admission, median (IQR)	8 (5–10)	7 (5–10)	0.011	8 (5.5–10)	8 (7–10)	0.648
BMI	29.8 (26.2–32.4)	29.6 (26.9–33.5)	0.521	29.8 (26.2–32.4)	29.6 (27–33.1)	0.939
Smoking behaviour			0.578			0.886
Never smoker	131 (69.3)	321 (65.1)		131 (69.3)	134 (67.3)	
Former smoker	49 (25.9)	144 (29.2)		49 (25.9)	56 (28.1)	
Current smoker	9 (4.8)	28 (5.7)		9 (4.8)	9 (4.5)	
Degree of dependency			0.874			0.366
None or mild	174 (92.1)	451 (91.5)		174 (92.1)	182 (91.5)	
Moderate	9 (4.8)	28 (5.7)		9 (4.8)	14 (7)	
Severe	6 (3.2)	14 (2.8)		6 (3.2)	3 (1.5)	
Arterial hypertension	97 (51.3)	279 (56.6)	0.216	97 (51.3)	97 (48.7)	0.612
Dyslipidemia	86 (45.5)	222 (45)	0.912	86 (45.5)	93 (43.7)	0.808
Diabetes mellitus	41 (21.7)	115 (23.3)	0.649	41 (21.7)	42 (21.1)	0.888
Ischaemic cardiopathy	10 (5.3)	35 (7.1)	0.395	10 (5.3)	12 (6)	0.753
Cerebrovascular disease	4 (2.1)	30 (6.1)	0.031	4 (2.1)	5 (2.5)	1
Dementia	7 (3.7)	20 (4.1)	0.832	7 (3.7)	11 (5.5)	0.393
Chronic heart failure	6 (3.2)	25 (5.1)	0.287	6 (3.2)	7 (3.5)	0.851
Chronic liver disease	8 (4.2)	12 (2.4)	0.213	8 (4.2)	7 (3.5)	0.715
Severe chronic renal failure	4 (2.1)	15 (3)	0.612	4 (2.1)	7 (3.5)	0.544
Cancer	9 (4.8)	29 (5.9)	0.568	9 (4.8)	6 (3)	0.372
COPD	10 (5.3)	39 (7.9)	0.236	10 (5.3)	13 (6.5)	0.605
Asthma	3 (1.6)	22 (4.5)	0.108	3 (1.6)	3 (1.5)	1
OSAS	11 (5.8)	43 (8.7)	0.209	11 (5.8)	15 (7.5)	0.499
Charlson index, median (IQR)	0 (0–1)	0 (0–2)	0.105	0 (0–1)	0 (0–1)	0.918

MP: methylprednisolone. DXM: dexamethasone. BMI: body mass index. IQR: interquartile range. COPD: chronic obstructive pulmonary disease. OSAS: obstructive sleep apnea syndrome. Severe chronic renal failure: Creatinine > 300 mg/dL or dialysis.

**Table 2 jcm-10-04465-t002:** Comparison of symptoms and signs upon admission between groups.

	Whole Cohort			Matched Sample		
	3-Day MP	DXM	*p*-Value	3-Day MP	DXM	*p*-Value
Cough, *n* (%)	149 (78.8)	344 (69.8)	0.018	149 (78.8)	142 (71.4)	0.089
Arthromyalgias, *n* (%)	57 (30.2)	122 (24.7)	0.150	57 (30.2)	49 (24.6)	0.221
Ageusia, *n* (%)	22 (11.6)	87 (17.6)	0.055	22 (11.6)	37 (18.6)	0.057
Anosmia, *n* (%)	19 (10.1)	78 (15.8)	0.054	19 (10.1)	37 (18.6)	0.017
Sore throat, *n* (%)	10 (5.3)	39 (7.9)	0.236	10 (5.3)	20 (10.1)	0.079
Headache, *n* (%)	22 (11.6)	81 (16.4)	0.118	22 (11.6)	36 (18.1)	0.075
Fever, *n* (%)	167 (88.4)	390 (79.1)	0.005	167 (88.4)	155 (77.9)	0.006
Dyspnea, *n* (%)	127 (67.2)	332 (67.3)	0.971	127 (67.2)	137 (68.8)	0.728
Diarrhea, *n* (%)	63 (33.3)	152 (30.8)	0.529	63 (33.3)	71 (35.7)	0.627
Vomiting, *n* (%)	11 (5.8)	26 (5.3)	0.778	11 (5.8)	9 (4.5)	0.563
Abdominal pain, *n* (%)	4 (2.1)	21 (4.3)	0.255	4 (2.1)	8 (4)	0.382
Heart rate, bpm median (IQR)	93 (82–105)	90 (80–102.5)	0.100	93 (82–105)	90 (80–105)	0.262
Respiratory rate > 20 bpm, *n* (%)	116 (61.4)	293 (59.4)	0.643	116 (61.4)	120 (60.3)	0828

MP: methylprednisolone. DXM: dexamethasone. IQR: interquartile range.

**Table 3 jcm-10-04465-t003:** Comparison of laboratory tests upon admission between groups.

	Whole Cohort			Matched Sample		
	3-Day MP	DXM	*p*-Value	3-Day MP	DXM	*p*-Value
PaO_2_/FiO_2_	271.4 (206–323.6)	304.8 (242.9–345.7)	<0.001	271.4 (206–323.6)	290.6 (222.1–338.1)	0.107
Lymphocytes × 10^6^/L, median (IQR)	820 (600–1090)	880 (630–1220)	0.026	820 (600–1090)	880 (630–1190)	0.081
CRP mg/L, median (IQR)	126.4 (69.8–222.1)	97.5 (56.1–162.8)	<0.001	126.4 (69.8–222.1)	106 (64.7–177)	0.039
LDH U/L, median (IQR)	393 (308.5–481)	339 (266.5–438.8)	<0.001	393 (308.5–481)	371 (282–459)	0.052
Ferritin mcg/L, median (IQR)	1440 (712.3–2110.8)	958.5 (454.9–1736.7)	<0.001	1440 (712.3–2110.8)	1217.2 (573–1862.7)	0.083
D-dimer ng/mL, median (IQR)	414 (250–828.5)	339 (250–651.5)	0.016	414 (250–828.5)	388 (250–804)	0.246

MP: methylprednisolone. DXM: dexamethasone. CRP: C-reactive protein. LDH: lactate dehydrogenase. IQR: interquartile range.

**Table 4 jcm-10-04465-t004:** Comparison of additional treatments between groups.

	Whole Cohort			Matched Sample		
	3-Day MP	DXM	*p*-Value	3-Day MP	DXM	*p*-Value
Remdesivir	3 (1.6)	177 (35.9)	<0.001	3 (1.6)	8 (4)	0.222
Tocilizumab	110 (58.2)	89 (18.1)	<0.001	110 (58.2)	55 (27.6)	<0.001
LMWH			<0.001			0.001
No Low doses Intermediate doses Full doses	6 (3.2) 106 (56.1) 36 (19) 41 (21.7)	53 (10.8) 288 (58.4) 96 (19.5) 56 (11.4)		6 (3.2) 106 (56.1) 36 (19) 41 (21.7)	18 (9) 111 (55.8) 51 (25.6) 19 (9.5)	

MP: methylprednisolone. DXM: dexamethasone. LMWH: Low-molecular-weight heparin.

**Table 5 jcm-10-04465-t005:** Comparison of primary and secondary outcomes between groups.

	Whole Cohort			Matched Sample		
	3-Day MP	DXM	*p*-Value	3-Day MP	DXM	*p*-Value
Primary outcome *n* (%)						
In-hospital mortality	59 (31.2)	88 (17.8)	<0.001	59 (31.2)	35 (7.1)	0.002
Secondary outcomes *n* (%)						
HFNC	65 (34.4)	186 (37.7)	0.419	65 (34.4)	84 (42.2)	0.113
NIMV	50 (26.5)	121 (24.5)	0.606	50 (26.5)	50 (25.1)	0.765
IMV	49 (25.9)	68 (13.8)	<0.001	49 (25.9)	29 (14.6)	0.005
ICU admission	55 (29.1)	101 (20.5)	0.017	55 (29.1)	37 (18.6)	0.015
Composite variable	118 (62.4)	225 (45.6)	<0.001	118 (62.4)	100 (50.3)	0.016
Length of stay (days), median (IQR)	14 (8–24.5)	11 (7–18.5)	<0.001	14 (8–24.5)	11 (7–19)	0.001

MP: methylprednisolone. DXM: dexamethasone. HFNC: high Flow nasal cannula. NIMV: non-invasive mechanical ventilation. IMV: invasive mechanical ventilation. ICU: intensive care unit.

**Table 6 jcm-10-04465-t006:** Risk factors for in-hospital mortality in the matched sample.

	Univariate Analysis		Multivariate Analysis	
	OR (95%CI)	*p*-Value	OR (95%CI)	*p*-Value
Age/year	1.06 (1.04–1.09)	<0.001	1.08 (1.05–1.10)	<0.001
Gender (female)	1.04 (0.62–1.75)	0.881	NS	
BMI	0.99 (0.95–1.04)	0.76		
Smoking behaviour			NS	
Never smoker	1 ref.			
Former smoker	1.56 (0.94–2.61)	0.088		
Current smoker	2.38 (0.88–6.41)	0.088		
Degree of dependency				
None or mild	1 ref.		1 ref.	
Moderate	2.29 (0.96–5.49)	0.063	1.37 (0.50–3.77)	0.544
Severe	12.47 (2.54–61.3)	0.002	8.77 (1.51–51.08)	0.016
Arterial hypertension	1.99 (1.23–3.20)	0.005	NS	
Dyslipidemia	1.54 (0.97–2.45)	0.071	NS	
Diabetes mellitus	1.71 (1.01–2.92)	0.048	NS	
Ischaemic cardiopathy	0.92 (0.33–2.55)	0.866		
Cerebrovascular disease	0.89 (0.18–4.37)	0.887		
Dementia	4.26 (1.63–11.13)	0.003	NS	
Chronic heart failure	2.01 (0.64–6.30)	0.232		
Chronic liver disease	0.78 (0.21–2.81)	0.697		
Severe chronic renal failure	1.82 (0.52–6.37)	0.347		
Cancer	2.88 (1.01–8.16)	0.047	NS	
COPD	2.57 (1.09–6.08)	0.031	NS	
Asthma	1.58 (0.28–8.75)	0.603		
OSAS	1.43 (0.60–3.40)	0.422		
PaO_2_/FiO_2_	0.99 (0.99–0.99)	<0.001	0.99 (0.99–0.99)	<0.001
Respiratory rate > 20 bpm	1.84 (1.11–3.04)	0.018	NS	
Lymphocytes/× 10^6^/L	1.00 (1.00–1.00)	0.66		
CRP/mg/L	1.01 (1.01–1.01)	0.001	NS	
LDH/U/L	1.01 (1.01–1.01)	<0.001	1.01 (1.01–1.01)	0.012
Ferritin/mcg/L	1.01 (1.01–1.01)	0.774		
D-dimer/ng/mL	1.01 (1.01–1.01)	0.101		
Corticosteroids				
DXM	1 ref.		1 ref.	
3-day MP	2.13 (1.32–3.43)	0.002	2.30 (1.33–3.98)	0.003
Remdesivir	0.31 (0.04–2.42)	0.261		
Tocilizumab	0.95 (0.59–1.51)	0.815		

BMI: body mass index. NS: Not significant. COPD: chronic obstructive pulmonary disease. OSAS: obstructive sleep apnea syndrome. CRP: C-reactive protein. LDH: lactate dehydrogenase. MP: methylprednisolone. DXM: dexamethasone.

## Data Availability

The datasets generated during and/or analyzed during the current study are available from the corresponding author on reasonable request.

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
