# Peer review of "High-Dose Methylprednisolone Pulses for 3 Days vs. Low-Dose Dexamethasone for 10 Days in Severe, Non-Critical COVID-19: A Retrospective Propensity Score Matched Analysis"

_jcm, 2021, doi:10.3390/jcm10194465_

Round 1

Reviewer 1 Report

The study on the use of high-dose methylprednisolone pulses for 3 days vs. low-dose 2 dexamethasone for 10 days in severe, non-critical COVID-19, examines an alternative to current guidelines regime, and comes at a time of another increase in Covid cases, therefore is timely and important.  The overall work is well performed and presented and I only have minor comments and questions for the authors.

  1. was the use of MP completely random? It seams to me the treating physicians selected MP for patients presenting with a more pronounced lab inflammatory profile. Please comment
  2. Please provide more details in the suppl on the schemes of MP used. 
  3. The presentation of previously published studies in the suppl. does not seam necessary to me
  4. In the suppl. please provide more details on the missing data
  5. In the suppl.please provide more details on the creation of the matched samples 
  6. Was the sample size adequate for the multivariate analysis in all the parameters examined
  7. Could you please provide data on the rate of secondary infections? it appears as a possible cause of the increased mortality
  8. Lab variables known to be associated with increased mortality such as inflammatory markers and d-dimers are not recognised as such in your analysis. Could it be related to sample size? what if specific cutoff values were used? 

Reviewer 2 Report

The publication raises very serious issues.  The concept of therapy for patients with SARS-Co is being developed today and each clinical report carries great significance. Every study and clinical experience can bring help in solving the problem and reducing the number of complications. Therefore, publications should be based on maximum informed and relevant observations.
Please complete the methodology for selecting therapy for a patient treated with DXM or MP.
From clinical diagnostics in compared groups we find only PaO2/FiO2, no further clinical signs of respiratory failure PaCO2, respiratory count, no other diagnostic tests - reference to imaging diagnostics and comparison of baseline changes in lung imaging studies.
In laboratory tests, the MP group had significantly higher levels of CRP (<0.03) and fever in patients (p<0.00).
While the statistical analysis is valid, the omission of these differences from the baseline criteria is questionable.
